



# Revisiting paramagnetic relaxation enhancements in slowly rotating systems: how long is the long range?

**Giovanni Bellomo**[1,2,a], **Enrico Ravera**[1,2], **Vito Calderone**[1,2], **Mauro Botta**[3], **Marco Fragai**[1,2],
**Giacomo Parigi**[1,2], **and Claudio Luchinat**[1,2]

[1]Magnetic Resonance Center (CERM) and Department of Chemistry,
University of Florence, via Sacconi 6, Sesto Fiorentino, Italy
[2]Consorzio Interuniversitario Risonanze Magnetiche di Metalloproteine (CIRMMP), Sesto Fiorentino, Italy
[3]Dipartimento di Scienze e Innovazione Tecnologica, Università del Piemonte Orientale "Amedeo Avogadro",
Viale T. Michel 11, 15121 Alessandria, Italy
[a]present address: Laboratory of Clinical Neurochemistry, Neurology Clinic, University of Perugia,
Piazzale Lucio Severi 1/8, 06132 Perugia (PG), Italy

**Correspondence:** Claudio Luchinat (luchinat@cerm.unifi.it)

**Abstract.** Cross-relaxation terms in paramagnetic systems that reorient rigidly with slow tumbling times can increase the effective longitudinal relaxation rates of protons of more than 1 order of magnitude. This is evaluated by simulating the time evolution of the nuclear magnetization using a complete relaxation rate-matrix approach. The calculations show that the Solomon dependence of the paramagnetic relaxation rates on the metal–proton distance (as $r^{-6}$) can be incorrect for protons farther than 15 Å from the metal and thus can cause sizable errors in $R_1$-derived distance restraints used, for instance, for protein structure determination. Furthermore, the chemical exchange of these protons with bulk water protons can enhance the relaxation rate of the solvent protons by far more than expected from the paramagnetic Solomon equation. Therefore, it may contribute significantly to the water proton relaxation rates measured at magnetic resonance imaging (MRI) magnetic fields in the presence of slow-rotating nanoparticles containing paramagnetic ions and a large number of exchangeable surface protons.

## 1 Introduction

Paramagnetic relaxation rates are largely applied for macromolecular structure determination, because they provide information on the distance of the macromolecule nuclei from the paramagnetic metal ion as well as in the field of magnetic resonance imaging (MRI) (Bertini et al., 2017). Image contrast in MRI is in fact largely determined by the different nuclear relaxation rates of the water protons present in the different tissues of the human body (Koenig and Brown, 1990). However, in many cases the intrinsically low difference among relaxation rates of water protons in different tissues requires the use of contrast agents to highlight the presence of pathological conditions (Aime et al., 2006, 2019; Wahsner et al., 2019). In this study, we explore the possibil-

ity of increasing the efficacy of a paramagnetic molecule as an MRI contrast agent by exploiting cross-relaxation effects.

The relaxation rate is defined assuming a monoexponential time dependence of the magnetization during the recovery of its equilibrium conditions after a perturbation. However, the presence of dipole–dipole coupled nuclear spins can result in magnetization time dependences which are not monoexponential (Banci and Luchinat, 1998; Neuhaus and Williamson, 1989; Solomon, 1955). Indeed, in the presence of multiple nuclei, the dipole–dipole coupling between the spins can cause exchange of magnetization from one to another, and this effect can propagate diffusively throughout the macromolecule (spin diffusion). This is a well-known, although often overlooked, feature, which can be correctly taken into account by complete relaxation rate-matrix analy-

sis (Boelens et al., 1989; Olejniczak et al., 1986; Post et al., 1990) through programs like CORMA (Borgias et al., 1989).

In the presence of unpaired electron(s) (e.g., radicals or paramagnetic metal complexes), the dominant dipole–dipole interaction for nuclear spins is often that with the spin of the unpaired electron(s), even if the latter is much farther than other neighboring nuclei. In this case, the corresponding paramagnetic relaxation rate constant is described by the Solomon equation for paramagnetic solutions (Solomon, 1955), which dictates a dependence of the paramagnetic relaxation rate on the inverse sixth power of the distance of the nuclear spin from the paramagnetic center ($r^{-6}$). However, the presence of multiple dipole–dipole interactions between nuclei close to one another is expected to considerably increase the nuclear relaxation rate. Hereafter, we call the Solomon equation the widely used equation provided by Solomon for paramagnetic solutions (Solomon, 1955), although in the same work Solomon also provided the coupled equations which include cross-relaxation terms and should be taken into account for treating the case of nucleus dipole–nucleus dipole interacting spins.

We have here modified program CORMA to calculate (i) the longitudinal relaxation rates of protons in molecules with known structure, in the presence of paramagnetic ions, taking into account all cross-relaxation effects, and (ii) the longitudinal relaxation rates of the bulk water protons, in the presence of some protons of the molecule in exchange with the bulk (Libralesso et al., 2005; Ravera et al., 2013). This model allowed us to calculate the deviations of the relaxation enhancements with respect to the values predicted by the paramagnetic Solomon equation on the basis of the metal–proton distances and thus the gain in relaxation rate values due to the network of the dipole–dipole interactions.

## 2    Complete relaxation matrix analysis

If a macromolecule is dissolved in solution, the longitudinal relaxation rate of the solvent nuclei increases with respect to the value of the pure solvent molecules due to the presence, at the surface of the macromolecule, of solvent molecules in chemical exchange with bulk solvent molecules. The correlation time for the dipole–dipole interactions involving these solvent molecule nuclei is the shortest between the reorientation time of the macromolecule ($\tau_R$) and their lifetime ($\tau_{M,i}$).

Using the complete relaxation rate-matrix approach (Borgias et al., 1989; Jayalakshmi and Rama Krishna, 2002) (see Supplement), the relaxation rates of the solvent molecule nuclei interacting with the macromolecule and of the bulk solvent molecule nuclei can be calculated by including in the relaxation matrix as many extra rows and columns as the number of nuclei belonging to the interacting solvent molecules and an additional row and column relative to bulk solvent nuclei. Assuming $M$ solvent nuclei interacting with the macromolecule (composed of $N$ nuclei), and using a "normalized"

magnetization for the bulk nuclei,

$$
\boldsymbol{M}' = \begin{pmatrix} M_z^I \\ \vdots \\ f M_z^B \end{pmatrix} = \begin{pmatrix} M_z^I \\ \vdots \\ \tilde{M}_z^B \end{pmatrix},
$$

the relaxation matrix is represented by Eq. (1) TS1 (see Supplement) where $k_i = (\tau_{M,i})^{-1}$ are the exchange rate constants, $f$ is the ratio between the macromolecular concentration and the solvent molecule nuclei concentration, $\rho_i$ and $\sigma_{ij}$ are the (diamagnetic) auto and cross-relaxation rates (see Supplement), and $\rho_B$ is the relaxation rate of bulk solvent nuclei in the absence of the macromolecule. In order to consider the contribution to relaxation caused by a paramagnetic metal ion present in the macromolecule, the terms $R_{1M,i}$ appear in the diagonal elements of the relaxation matrix. They correspond to the Solomon paramagnetic relaxation rates TS2

$$
R_{1M,i} = \frac{2}{15} \left( \frac{\mu_0}{4\pi} \right)^2 \frac{\gamma_I^2 g_e^2 \mu_B^2 S(S+1)}{r_{iM}^6}
$$
$$
\left[ \frac{7\tau_{ci}}{1 + \omega_S^2 \tau_{ci}^2} + \frac{3\tau_{ci}}{1 + 4\omega_I^2 \tau_{ci}^2} \right], \tag{2}
$$

where $S$ is the electron spin quantum number, $\omega_S^2 = g_e^2 \mu_B^2 B_0^2$, $\omega_I^2 = \gamma_I^2 B_0^2$, the correlation time is given by $\tau_{ci}^{-1} = \tau_R^{-1} + \tau_e^{-1} + \tau_{Mi}^{-1}$, $\tau_e$ is the electron relaxation time, $g_e$ is the electron $g$ factor, $\mu_B$ is the electron Bohr magneton, $\gamma_I$ is the nuclear magnetogyric ratio, and $B_0$ is the applied magnetic field. The relaxation matrix in Eq. (1) is not symmetric. However, we can define a symmetric matrix $\mathbf{R}_s = \mathbf{F}^{-1} \cdot \mathbf{R}' \cdot \mathbf{F}$, where

$$
\mathbf{F} = \begin{pmatrix} 1 & 0 & \dots & 0 \\ 0 & 1 & \dots & 0 \\ \vdots & \vdots & \ddots & \vdots \\ 0 & 0 & \dots & \sqrt{f} \end{pmatrix}. \tag{3}
$$

If $\lambda_s$ is the diagonal matrix of the eigenvalues and $\chi_s$ is the unitary eigenvector matrix of $\mathbf{R}_s$, the time evolution of the longitudinal magnetization is

$$
\boldsymbol{M}'(t) - \boldsymbol{M}'_{eq} = \mathbf{F} \cdot \chi_s \cdot \exp(-\lambda_s t) \cdot \chi_s^{-1} \cdot \mathbf{F}^{-1}
$$
$$
\cdot \left( \boldsymbol{M}'(0) - \boldsymbol{M}'_{eq} \right). \tag{4}
$$

## 3    Results and discussion

The relaxation rates of all protons belonging to a macromolecule containing a paramagnetic metal ion can be calculated using a modified version of program CORMA (Borgias et al., 1989), called CORMA-PODS (COmplete Relaxation Matrix Analysis – Paramagnetic Or Diamagnetic Solutions). In summary, after diagonalization of the relaxation matrix in Eq. (1), the time dependence of the longitudinal magnetization of all macromolecule protons as well as of bulk water

Magn. Reson., 2, 1–7, 2021

https://doi.org/10.5194/mr-2-1-2021

$$\mathbf{R'} = \begin{pmatrix} \rho_1 + k_1 + R_{1M,1} & \sigma_{12} & \cdots & \sigma_{1N} & \sigma_{1(N+1)} & \cdots & \sigma_{1(N+M)} & -k_1 \\ \sigma_{12} & \rho_2 + k_2 + R_{1M,2} & \cdots & \sigma_{2N} & \sigma_{2(N+1)} & \cdots & \sigma_{2(N+M)} & -k_2 \\ \vdots & \vdots & \ddots & \vdots & \vdots & \ddots & \vdots & \vdots \\ \sigma_{1N} & \sigma_{2N} & \cdots & \rho_N + k_N + R_{1M,N} & \sigma_{N(N+1)} & \cdots & \sigma_{N(N+M)} & -k_N \\ \sigma_{1(N+1)} & \sigma_{2(N+1)} & \cdots & \sigma_{N(N+1)} & \rho_{N+1} + k_{N+1} + R_{1M,N+1} & \cdots & \sigma_{(N+1)(N+M)} & -k_{N+1} \\ \vdots & \vdots & \ddots & \vdots & \vdots & \ddots & \vdots & \vdots \\ \sigma_{1(N+M)} & \sigma_{2(N+M)} & \cdots & \sigma_{N(N+M)} & \sigma_{(N+1)(N+M)} & \cdots & \rho_{N+M} + k_{N+M} + R_{1M,N+M} & -k_{N+M} \\ -fk_1 & -fk_2 & \cdots & -fk_N & -fk_{N+1} & \cdots & -fk_{N+M} & \rho_B + f\sum_i k_i \end{pmatrix}, \quad (1)$$

protons can be calculated from Eq. (4) (with all elements of the vector $\mathbf{M'}_{eq}$ equal to 1), with all elements of the vector $\mathbf{M'}(0)$ describing the initial longitudinal magnetizations and equal to the same value ($-1$ or 0 to simulate an inversion recovery or a 90° pulse, respectively), assuming that a non-selective radiofrequency pulse is applied. Although for some nuclei the magnetization recovery curves deviate from monoexponential functions as expected (see below), the "apparent" relaxation rates were calculated for simplicity as the rate constants of the assumed monoexponential time dependences of the magnetization curves.

### 3.1 Paramagnetic relaxation rates in high field NMR spectroscopy

We first checked whether cross-relaxation effects can cause sizable deviations of the nuclear relaxation rates from the expected $r^{-6}$ dependence predicted by the Solomon equation in paramagnetic proteins at high magnetic fields. Since the experimental rates are used to back-calculate, through the Solomon equation, the nucleus–metal distances to be employed as restraints for molecular structure determination, this would result in incorrect structural restraints.

A deviation between the correct metal–proton distances and those determined from the longitudinal relaxation rates was first experimentally observed by Led and coworkers (Ma et al., 2000) at 500 MHz for the protein plastocyanin, a copper(II) protein with a reorientation time of 6.2 ns and an electron relaxation time of 0.17 ns. Figure 1a shows the apparent paramagnetic relaxation enhancement of all plastocyanin protons in these conditions, calculated as the difference between the apparent rates obtained with including the paramagnetic metal and without it. The deviations from the Solomon behavior for many protons at distances larger than 15 Å result in metal–proton distances (Fig. 1b) somewhat smaller than the correct ones, in accordance with the experimental data. As already noted by Led, the experimental data tend to deviate more than predicted from CORMA. These deviations from the Solomon equation in the calculated data are due to both the different contributions from nucleus dipole–nucleus dipole interactions arising in the presence of the paramagnetic metal and cross-relaxation effects. A few protons at intermediate distances may also have slightly slower relaxation rates than expected from the Solomon equation. This is caused by the nucleus dipole–nucleus dipole interac-

tions among protons at large distances from the paramagnetic metal, which cause "magnetization losses" from the closer to the farther protons, with clear deviations from a monoexponential magnetization recovery (see Fig. S1 in the Supplement). Analogous behavior is calculated for the catalytic domain of the protein matrix metalloproteinase 12 (Balayssac et al., 2008; Benda et al., 2016) by replacing the catalytic zinc ion with high-spin cobalt(II), copper(II), or gadolinium(III) (Fig. S2), although the electron relaxation rates of the different metals differ by orders of magnitude. These deviations from the Solomon behavior are largely reduced or completely removed for amide and hydroxyl protons in perdeuterated conditions and for (isoleucine, leucine, and valine) methyl protons if the rest of the protein is perdeuterated (Fig. S3).

This analysis represents a warning against the use of distance restraints for the structural refinement of macromolecules, derived from experimental $R_1$ using the Solomon equation, for protons at distances farther than 15 Å from the paramagnetic metal. On the other hand, these calculations are performed on the assumption of completely rigid molecules (except methyl jumps), which is clearly an unrealistic assumption for biomolecules. Internal mobility may actually reduce the deviations with respect to the Solomon predictions. Fast local mobility is in fact of paramount importance in determining the relaxation rates. If methyl protons were fixed, instead of jumping fast between different positions, the deviations from the Solomon behavior for protons at distances larger than 20 Å were in fact significantly larger (see Fig. 1c, d).

### 3.2 Solvent water proton relaxation enhancement

The enhancement in nuclear relaxation calculated for protons at large distances from the paramagnetic metal can have important consequences also for the relaxation rate of solvent water protons in solutions containing paramagnetic macromolecules. As a test system, a synthetic model was used mimicking a sphere of hydrogen-bonded water molecules (arranged as in crystalline ice), with a gadolinium(III) ion in the center. In this model, each proton has another proton at about 1.5 Å and 8 protons between 2.5 and 3.1 Å, for a total of 844 protons. The electron relaxation of gadolinium is calculated assuming typical values for the electron relaxation parameters, $\Delta_t = 0.030\,\text{cm}^{-1}$ and $\tau_v = 20\,\text{ps}$ (Caravan et al.,

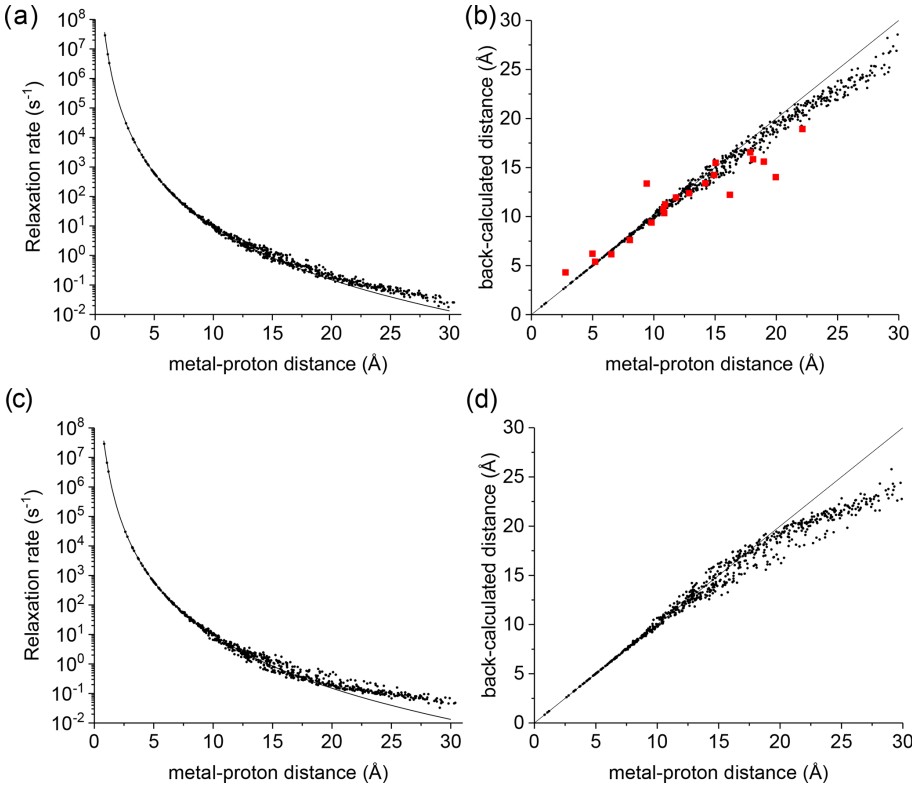

**Figure 1. (a)** Apparent paramagnetic relaxation rates calculated at 500 MHz for $Cu^{2+}$-plastocyanin protons. The line indicates the rates predicted with the Solomon equation. **(b)** Agreement between metal–proton distances as measured in the PDB 2GIM structure and back-calculated from the predicted $R_1$. **(c, d)** The same calculations are performed assuming fixed positions for methyl protons. The red points in panel **(b)** refer to the distances back-calculated from the experimental data (Ma et al., 2000).

1999; Li et al., 2002; Mastarone et al., 2011), which provide electron relaxation times of 4.2 ns at 1 T (Tesla) and 36 ns at 3 T.

Figure S4 shows the magnetization recovery curves for protons at different distances from the gadolinium(III) ion after a 90° pulse; the figure shows that for some nuclei there can be a deviation from a monoexponential function of time. The relaxation rates can however be calculated as rate constants of the monoexponential time dependence of the magnetization. These rates were first evaluated at 1 and 3 T for reorientation times of 50, 500, and 5000 ns and for all protons within the sphere, in the absence of chemical exchange with bulk water molecules (Fig. 2). Figure 2 also shows how the relaxation rates of protons relatively far from the metal increase with respect to the rates calculated from the Solomon equation (Eq. 2). This effect is of increasing importance when increasing the reorientation time of the molecule, the magnetic field (from 1 to 3 T), and the electron relaxation time (Fig. S5). In the absence of chemical exchange, and neglecting outer-sphere relaxation mechanisms (Freed, 1978), the relaxation rate of bulk water protons does not change with respect to the intrinsic water molecule relaxation value, $\rho_B$, fixed to 0.3 s$^{-1}$.

The effect of this relaxation enhancement on the solvent water proton $R_1$ was then evaluated in the presence of 100 superficial protons with an exchange rate of $10^4$ s$^{-1}$. The molar ratio $f$ between the macromolecular concentration and the solvent water proton concentration is assumed equal to $9 \times 10^{-6}$, corresponding to a macromolecular concentration of 0.001 mol dm$^{-3}$. The presence of these exchangeable protons causes a relaxation enhancement of bulk water protons, shown in Fig. 3a. This enhancement increases when increasing the reorientation time of the macromolecule, and for reorientation times of microseconds or larger it largely exceeds the paramagnetic enhancement calculated with the Solomon equation and 100 protons at the same distance from the gadolinium ion and with the same exchange rate. Of note, for such large reorientation times (and lack of any internal mobility), the paramagnetic enhancement can almost reach the values achieved at the same fields by small complexes with a water molecule coordinated to the gadolinium ion and used in MRI (as Gd-DOTA or Gd-DTPA) (Anelli et al., 2000; Caravan et al., 1999; Fragai et al., 2019).

Figure 3b shows the dependence of the bulk water proton relaxation rate on the exchange rate of the 100 exchangeable surface protons. Sizable paramagnetic enhancements can be achieved for exchange times shorter than milliseconds; the

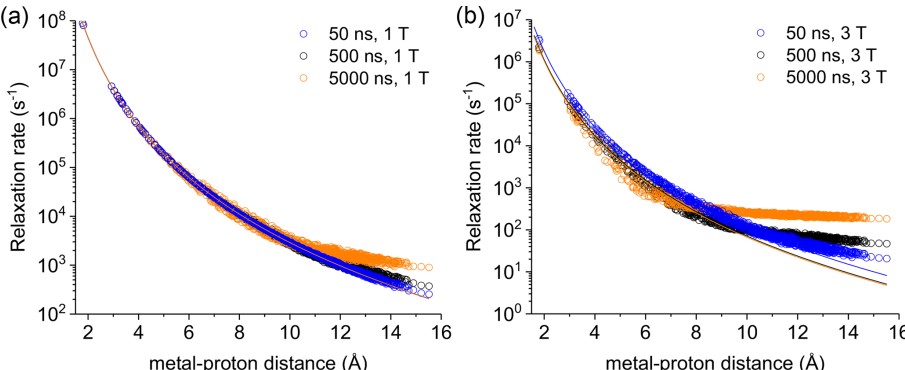

**Figure 2.** Apparent relaxation rates calculated at 1 T **(a)** and 3 T **(b)** for protons at different distances from a $Gd^{3+}$ ion in the macromolecular model with reorientation times of 50, 500, or 5000 ns. The lines indicate the Solomon relaxation rates calculated for the same reorientation times (colored accordingly).

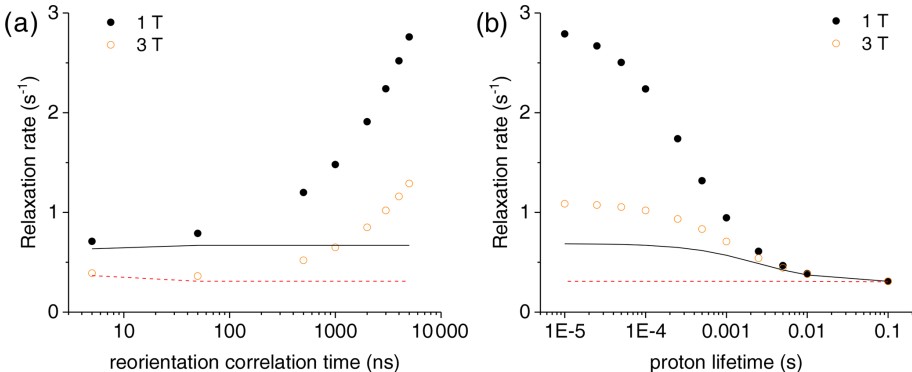

**Figure 3. (a)** Bulk water proton relaxation rates calculated at 1 and 3 T as a function of the reorientation time of the $Gd^{3+}$-containing macromolecular model (at 0.001 mol dm$^{-3}$ concentration), with 100 surface protons with an exchange rate of 0.1 ms. **(b)** Bulk water proton relaxation rates calculated at 1 and 3 T as a function of the exchange rate of 100 surface protons in the macromolecular model with a reorientation time of 3000 ns. The bulk water proton relaxation rates calculated with the Solomon equation, and according to $R_{1\text{bulk}} = R_{1\text{dia}} + f\left(R_{1M}^{-1} + \tau_M\right)^{-1}$, at 1 and 3 T are shown as solid and dashed lines, respectively. In all calculations, an intrinsic diamagnetic rate of 0.3 s$^{-1}$ is assumed.

enhancement increases with decreasing exchange time until a value on the order of the macromolecular reorientation time is reached.

The same model was used to evaluate the bulk water proton relaxation enhancement obtained in the presence of paramagnetic metal ions other than gadolinium. Interestingly, the effect is similar even for $S = 1/2$ ions with, e.g., an electron relaxation time of 4 ns, i.e., on the order of magnitude typical of type 2 copper(II). Indeed, the paramagnetic relaxation enhancements are about halved at 1 T but very similar to those calculated for $Gd^{3+}$ at 3 T (see Fig. S6).

In a second synthetic model, a gadolinium(III) ion is placed in the center of a sphere with six protons in octahedral geometry at a distance of 2.5 Å from the metal, each proton having further another six protons in octahedral geometry at the same distance, and so on. The farther protons are at a distance of 20 Å from the metal. One hundred protons on the surface of the sphere are assumed exchangeable with

an exchange rate of 0.1 ms. The relaxation enhancement of bulk water protons again increases significantly for reorientation times longer than microseconds, exceeding by about a factor 10 the paramagnetic enhancement calculated with the Solomon equation due to the same 100 protons at the same distance from the gadolinium ion and with the same exchange rate (Fig. S7).

## 4 Conclusions

The calculations performed indicate that the magnetization transfer from protons in a polymer matrix to water protons may provide valuable contributions to the water proton relaxation rates in the presence of a paramagnetic metal ion entrapped within the polymer (Rammohan et al., 2016; Ravera et al., 2020; Rotz et al., 2015). This contribution occurs when a paramagnetic metal ion is bound to a rigid macro-

molecule, composed of a network of hydrogen nuclei a few Å away from one another, with microsecond tumbling time and with hundreds of nuclei in the external layer in relatively fast exchange (tens to hundreds of microseconds) with bulk water protons. These conditions seem hard to meet experimentally, so that this effect cannot be easily exploited to increase the effectiveness and the safety of an MRI contrast agent. Nevertheless, it might prove useful when the paramagnetic ions are entrapped in slow-rotating proton-rich nanoparticles with sponge-like structures, allowing a large number of exchangeable surface protons, like Gd-based mesoporous silica nanoparticles (Carniato et al., 2018).

Importantly, these calculations also show that assuming a metal–proton distance dependence as $r^{-6}$ for the longitudinal relaxation rates of protons at more than 15 Å from the metal in a macromolecule can cause sizable errors. This should be taken into account when $R_1$-derived distance restraints are used in structural determination procedures.

**Code availability.** Calculations with CORMA-PODS will be soon possible through http://cormapods.cerm.unifi.it TS3.

**Data availability.** No data sets were used in this article. TS4

**Supplement.** The supplement related to this article is available online at: https://doi.org/10.5194/mr-2-1-2021-supplement.

**Author contributions.** CL conceived the project and CL, GP, ER, MF, and MB discussed all theoretical aspects and simulations. GB and GP developed the model code and GP, GB, ER, and VC performed the simulations. GP prepared the manuscript with contributions from all the co-authors.

**Competing interests.** The authors declare that they have no conflict of interest.

**Special issue statement.** This article is part of the special issue "Robert Kaptein Festschrift". It is not associated with a conference.

**Acknowledgements.** Claudio Luchinat warmly remembers the many friendly discussions over the years with the late Jens J. Led. The work was carried out within the framework of the COST CA15209 Action ("European Network on NMR Relaxometry").

**Financial support.** This research has been supported by the Ministero dell'Istruzione, dell'Università e della Ricerca (grant no. PRIN 2017A2KEPL "Rationally designed nanogels embedding paramagnetic ions as MRI probes"), the Ministero della Salute (grant no. GR-2016-02361586), the European Commission through H2020 FET-Open project HIRES-MULTIDYN (grant agreement no. 899683), and the Fondazione Cassa di Risparmio di Firenze.

**Review statement.** This paper was edited by Rolf Boelens and reviewed by Fabien Ferrage, Carlos Geraldes, and Carol Post.

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

## Remarks from the typesetter