# Peer review of "Revisiting paramagnetic relaxation enhancements in slowly rotating systems: how long is the long range?"

_Magnetic Resonance, 2020_

## Referee Comment (RC1) · Carlos Geraldes (Referee) · 11 Dec 2020

The present study is concise, technically correct and very useful for future correct use of paramagnetic relaxation rates to derive distance restraints in the structural determination of macromlecular systems procedures, as the usual distance dependence from the Solomon equations may not be obeyed at large diatances due to cross relaxatio effects. It is also useful to the design of MRI contrast agents, as the chemical exchange of surface protons in slowly rotating macromolecular with the bulk may enhance the relaxation rates obtained.

[Discussion paper]

---

## Short Comment (SC1) · 20 Dec 2020

The article identifies an important mechanism by which the best R1 relaxation rate measurements will yield unexpectedly large PREs, as the overall relaxation rate is enhanced by cross-relaxation between protons that themselves experience PREs. The effect becomes quite significant for greater distances from the paramagnetic centre. Interestingly, at intermediate distances the opposite effect can occur too, i.e. the total PRE can become slower than expected based on the simple Solomon equation. Is this due to NOE contacts with protons located at greater distance from the paramagnetic centre?

Can the authors predict what the situation would be like for backbone amide protons in a perdeuterated protein in H2O, where the only protons are the amide protons from the backbone and side chains (also allowing the presence of hydroxyl protons)? What would the situation be like if the methyls of isoleucine, valine and leucine are protonated whereas the rest of the protein is perdeuterated?

Line 136: exchange rates usually are expressed in s-1, not seconds.

---

## Author Comment (AC1) · 22 Dec 2020

Thank you very much for your positive comments. We are happy that you like the manuscript.

---

## Author Comment (AC2) · 22 Dec 2020

We are grateful to Gottfried Otting for his constructive comments, which helped us to make the message clearer and more complete. Here below please find our answers. The manuscript will be modified accordingly.

Comment 1. Interestingly, at intermediate distances the opposite effect can occur too, i.e. the total PRE can become slower than expected based on the simple Solomon equation. Is this due to NOE contacts with protons located at greater distance from the paramagnetic centre?

Answer: As noticed, at intermediate distances the total PRE can actually become slightly slower than expected from the Solomon equation. This is indeed caused by the dipole-dipole interactions with protons at larger distances from the paramagnetic metal, which cause "magnetization losses" from the closer to the farther protons. This is clarified by an additional simple simulation where 6 protons are placed along a straight line at 10, 12, 14, 16, 18 and 20 Å from a gadolinium ion. The results of the simulation are shown in the figure S6 below. The magnetization of the proton at 10 Å recovers its equilibrium value slower than predicted from an exponential behavior, so that the monoexponential fit provides a longer relaxation time. The figure also shows (black dotted lines) the magnetization curves expected from the Solomon equation for the two protons at 10 and 20 Å. Clearly the first points of the magnetization recovery for the proton at 10 Å agree with the relaxation rate predicted by the Solomon equation. This comment will be introduced in the revised version, and the figure S6 will be added in the Supplement.

Comment 2. Can the authors predict what the situation would be like for backbone amide protons in a perdeuterated protein in H2O, where the only protons are the amide protons from the backbone and side chains (also allowing the presence of hydroxyl protons)? What would the situation be like if the methyls of isoleucine, valine and leucine are protonated whereas the rest of the protein is perdeuterated?

Answer: We have performed the suggested calculations. As shown in Fig. S7A, amide and hydroxyl protons in perdeuterated conditions almost recover the rates predicted by the Solomon equation, although a slight smaller distance can anyway be calculated for protons at more than 20 Å from the paramagnetic metal. Methyl protons fully recover the Solomon behavior if the rest of the protein is perdeuterated (Fig. S7C). These plots will be added to the Supplement of the revised version.

Comment 3. Line 136: exchange rates usually are expressed in s-1, not seconds. Answer: In the revised version we will write "with an exchange rate of $10^4$ s$^{-1}$". Thank you for having pointed this out.

[Figure]

[Figure]

[Figure]

Figure S6. Calculated magnetization recovery for 6 protons placed along a straight line at 10, 12, 14, 16, 18 and 20 Å from a gadolinium ion, at 700 MHz (upper panel). The black dotted lines show the (monoexponential) behavior predicted from the Solomon equation for the two protons at 10 and 20 Å. The magnetization data calculated for the 6 protons are clearly not monoexponential. The monoexponential fits (solid colored lines) provide the relaxation rates and the back-calculated distances shown in the lower panels.

**Fig. 1.**

[Figure]

Figure S7. (A) Paramagnetic relaxation rates calculated at 500 MHz for $Cu^{2+}$-plastocyanin exchangeable (amide and hydroxyl) protons, in perdeuterated conditions. The line indicates the rates predicted with the Solomon equation. (C) Paramagnetic relaxation rates for isoleucine, leucine and valine methyl protons, assuming perdeuteration of all other hydrogens. (B and D) Agreement between metal-proton distances as measured in the PDB 2GIM structure and back-calculated from the predicted $R_1$ shown in panels A and C, respectively.

**Fig. 2.**

---

## Short Comment (SC2) · 23 Dec 2020

The CORMA program as modified by the authors clearly is very useful. Would it be possible to install the software in the NMRbox provided by Jeffrey Hoch to make it generally accessible? This would not only be helpful for others but also increase the impact and citation rate of the present article.

While the original CORMA software was developed many years ago in the group of Tom James at UCSF, it may be possible to obtain permission to make the new version accessible in this way?

---

## Referee Comment (RC2) · Fabien Ferrage (Referee) · 24 Dec 2020

This manuscript describes a theoretical investigation of the effect of nucleus-nucleus dipole-dipole interactions on longitudinal relaxation rates measured in a protein, as well as the same effect combined with chemical exchange of protons on the longitudinal relaxation of water protons in solutions of a paramagnetic complex, as encountered with contrast agents for MRI. The authors use the CORMA approach to include all dipole-dipole interactions between all nuclei in the molecule, including all relaxation pathways (but neglecting all cross-correlated cross-relaxation pathways). The authors show that nucleus-nucleus interactions cannot be neglected when the electron-nucleus

distance is larger than 15 Å. In the case of a paramagnetic complex, the authors show that chemical exchange of protons in the complex or weakly bound to the complex, can lead to enhancements of the longitudinal relaxation rate of water. Particularly when the complex tumbles slowly and when the exchange of magnetization is fast (a few ms or faster) This is a very interesting manuscript. The work is convincing and nicely presented. The mention of the late Jens Led in the acknowledgements is very nice given how closely related this work is to several studies from the Led group over the years. I list below a series of mostly technical points that should be addressed to further improve the manuscript and make it more accessible.

-It is not fully clear from the main text what the authors mean by "calculated rates". The authors use the adapted CORMA approach, they compute the evolution of the magnetization from Eq. 4. After this, one has to check Fig. S2 (not the main text) to find a fit of recovery to obtain the "calculated rate"? If so, are the recovery monoexponential? There seem to be some deviations in Figure S2.

-In the discussion of Figure 1, the authors should make it more clear whether the effect discussed is due to cross-relaxation with faster relaxing protons (a "selective T_1" effect) and not simply the contribution of nucleus dipole-nucleus dipole interactions to relaxation (a "non-selective T1" effect).

-In the light of the discussion between the authors and Gottfried Otting on a decrease of paramagnetic relaxation enhancements at intermediate distances, could this effect explain part of the surprising observations made by Flemming Hansen and Jens Led in their beautiful investigation of the blue copper site in Plastocyanin (JACS 2004)?

-The authors rightfully refer the contribution to relaxation from interactions with electrons to the seminal work of Solomon by mentioning the "Solomon Equation". However, I find this expression possibly confusing, especially in the context of this investigation. The legacy of Ionel Solomon reached beyond the expression of relaxation in paramagnetic systems and it is widely accepted in NMR that the Solomon EquationS describe

the evolution of magnetization in the presence of dipolar cross-relaxation, which is also perfectly relevant to this study. I believe that the authors should mention these Solomon Equations (introduced in the same Phys. Rev. 1955 article). It would be fantastic if the authors could take this opportunity to clarify the Solomon Equation vs. Solomon Equations issue, possibly by referring to the former as the Bloombergen-Solomon Equation as some do in the literature.

-I wonder if calculating the evolution of the magnetization would be easier with the use of the Homogeneous Master Equation (Levitt and di Bari 1922).

-On lines 128-129, the authors mention that, in the absence of exchange, the bulk water relaxation rate is not altered by the interaction with the electron magnetic dipole. Is it true or is it an approximation since outer-sphere relaxation mechanisms are not the topic of this investigation?

- I am not sure I understand exactly how the relaxation rates of the bulk calculated from the Solomon equation represented by dashed and solid lines in Figure 3 were calculated, what was included in each calculation.

-In the spirit of Magnetic Resonance, it would be preferable that the authors publish the simulation code used in their study.

Minor: -On lines 32-34, the question of non-monoexponential evolution of polarization in the presence of cross-relaxation could fill out volumes. The reference to the work of Banci and Luchinat is perfectly relevant here but could be accompanied by a general reference to nuclear Overhauser effects (e.g. the already cited Solomon article or the textbook by Neuhaus and Williamson). -Similarly, the "Furthermore" on line 33 could be replaced by "Indeed" or any other suggestion by the authors since cross-relaxation is the cause of non-monoexponentiality. -Including chemical exchange in the CORMA approach has been done in the past, for instance by Jayalakshmi and Rama Krishna (JMR 2002). -Many symbols are not defined in the main text. I may have missed some but could not find the definition of $\rho_i$, $k_i$, $\sigma_{ij}$, $g_e$, $\mu_B$,

B_0, gamma_i, tau_e, Delta_1, and tau_nu. -Unless this is a format requirement of Magnetic Resonance, I would suggest the authors use a first page with article title and authors list in the supplementary information document.

———————————————

---

## Author Comment (AC3) · 29 Dec 2020

We are glad that you like the manuscript and we thank you for your constructive comments. Here below a point-by-point answer:

-It is not fully clear from the main text what the authors mean by "calculated rates". The authors use the adapted CORMA approach, they compute the evolution of the magnetization from Eq. 4. After this, one has to check Fig. S2 (not the main text) to find a fit of recovery to obtain the "calculated rate"? If so, are the recovery monoexponential? There seem to be some deviations in Figure S2.

It is correct that the recovery is not monoexponential. However, the "calculated rates" were indeed obtained from a monoexponential fit of the magnetization curves. We will clarify this point immediately after the first paragraph of the Results and discussion section, by stating that "although for some nuclei the magnetization recovery curves deviate from monoexponential functions as expected (see below), the relaxation rates were calculated for simplicity as the rate constants of the monoexponential time-dependences of the magnetization curves".

-In the discussion of Figure 1, the authors should make it more clear whether the effect discussed is due to cross-relaxation with faster relaxing protons (a "selective T_1" effect) and not simply the contribution of nucleus dipole-nucleus dipole interactions to relaxation (a "non-selective T1" effect).

Both indicated effects are effective when a non-selective pulse is applied, as done in our simulations. Most of the deviations is due to the different contributions from nucleus dipole-nucleus dipole interactions caused by the hyperfine coupling, but also cross relaxation effects are present. This can be checked by comparing the calculated rates from selective and non-selective experiments (see Figure below). This will be clarified in the text.

-In the light of the discussion between the authors and Gottfried Otting on a decrease of paramagnetic relaxation enhancements at intermediate distances, could this effect explain part of the surprising observations made by Flemming Hansen and Jens Led in their beautiful investigation of the blue copper site in Plastocyanin (JACS 2004)?

Although this effect may slightly contribute to the experimental observations by Hansen and Led, we do not expect sizable contributions in that case. In fact, whereas the effects that we calculate are sizable at 13 Å or farther from the metal, the deviations were observed by Hansen and Led at less than 10 Å from the metal.

-The authors rightfully refer the contribution to relaxation from interactions with electrons to the seminal work of Solomon by mentioning the "Solomon Equation". However,

I find this expression possibly confusing, especially in the context of this investigation. The legacy of Ionel Solomon reached beyond the expression of relaxation in paramagnetic systems and it is widely accepted in NMR that the Solomon EquationS describe the evolution of magnetization in the presence of dipolar cross-relaxation, which is also perfectly relevant to this study. I believe that the authors should mention these Solomon Equations (introduced in the same Phys. Rev. 1955 article). It would be fantastic if the authors could take this opportunity to clarify the Solomon Equation vs. Solomon Equations issue, possibly by referring to the former as the Bloombergen-Solomon Equation as some do in the literature.

We will clarify that we call Solomon equation the widely used equation provided by Solomon for paramagnetic solutions (Solomon, 1955), although in the same work Solomon also provided the coupled equations which include cross-relaxation terms and should be taken into account for treating the case of nucleus dipole-nucleus dipole interacting spins. We prefer not to refer to the Bloembergen-Solomon equation because it includes contributions from Fermi-contact relaxation, which are not considered in this work.

-I wonder if calculating the evolution of the magnetization would be easier with the use of the Homogeneous Master Equation (Levitt and di Bari 1992).

This could be possible, but we have preferred a more "classical" approach, to better control all steps in the implementation of the model.

-On lines 128-129, the authors mention that, in the absence of exchange, the bulk water relaxation rate is not altered by the interaction with the electron magnetic dipole. Is it true or is it an approximation since outer-sphere relaxation mechanisms are not the topic of this investigation?

Contributions from outer-sphere relaxation are not considered in this work because of the large distance of the water molecules from the paramagnetic center, and thus not included into the model. This will be clarified in the text.

[Figure]

- I am not sure I understand exactly how the relaxation rates of the bulk calculated from the Solomon equation represented by dashed and solid lines in Figure 3 were calculated, what was included in each calculation.

The relaxation rates at 1 T (solid lines) and 3 T (dashed lines) are calculated from the Solomon relaxation rates $R\_1M$ of the exchangeable protons in the absence of any cross-relaxation terms, according to the relationship $R\_1bulk=R\_1dia+ f(R\_1M^{(-1)}+\tau\_M)^{(-1)}$. This will be clarified in the text.

-In the spirit of Magnetic Resonance, it would be preferable that the authors publish the simulation code used in their study.

We will contact the authors of CORMA and, if allowed, we will make available a version of the code as soon as completed with a user-friendly interface.

Minor:

-On lines 32-34, the question of non-monoexponential evolution of polarizationin the presence of cross-relaxation could fill out volumes. The reference to the work of Banci and Luchinat is perfectly relevant here but could be accompanied by a general reference to nuclear Overhauser effects (e.g. the already cited Solomon article or the textbook by Neuhaus and Williamson).

We agree to cite these more general references.

-Similarly, the "Furthermore" on line 33 could be replaced by "Indeed" or any other suggestion by the authors since cross-relaxation is the cause of non-monoexponentiality.

OK

-Including chemical exchange in the CORMA approach has been done in the past, for instance by Jayalakshmi and Rama Krishna (JMR 2002).

This work will be cited. Thank you for pointing it out.

-Many symbols are not defined in the main text. I may have missed some but could not find the definition of rho_i, k_i, sigma_ij, g_e, mu_B, B_0, gamma_i, tau_e, Delta_1, and tau_nu.

All symbols will be defined in the revised version of the manuscript.

-Unless this is a format requirement of Magnetic Resonance, I would suggest the authors use a first page with article title and authors list in the supplementary information document.

OK

Thank you for all your valuable comments.

[Figure]

[Figure]

Figure 1. Agreement between metal-proton distances in Cu$^{2+}$-plastocyanin as measured in the PDB 2GIM structure and back-calculated from the predicted $R_1$ at 500 MHz: left panel, the relaxation rates are predicted by simulating a non-selective experiment (same plot shown in Figure 1 of the manuscript), right panel, the relaxation rates are predicted by simulating a selective experiment.

**Fig. 1.**

[Figure]

---

## Author Comment (AC4) · 29 Dec 2020

We will contact the authors of CORMA and, if allowed, we will make available a version of the code as soon as completed with a user-friendly interface.

---

## Referee Comment (RC3) · Carol Post (Referee) · 5 Jan 2021

The authors present a thoughtful study to examine paramagnetic enhancement of nuclear relaxation accounting for multiple spin effects using a complete relaxation-rate matrix analysis. The results are interesting, clearly presented, and substantiate the two conclusions of the study: first, that paramagnetic metal ions cannot be exploited for altering water proton relaxation in MRI contrast agents because the needed conditions for reorientation time of the macromolecule and the exchange time for water solvents are not within typical values; second, that intermolecular distances greater than 15 angstroms estimated from paramagnetic relaxation enhancements appear shorter

than their actual values due to multiple-spin effects. This information provides important insight.

I have a few simple suggestions for the authors.

- Line 37: There were a number of groups who examined multiple-spin effects on cross-relaxation rates using a complete rate matrix at the same time as Borgias and James, 1989. It would be nice to reference some of the other papers as well, particularly one by Kaptein (e.g. Boelens and Kaptein, J Mag Res 1989; Olejniczak and Fesik J Mag Res 1986; co-workers and Gorenstein JACS 1990).

- Line 55 and elsewhere: The use of the phrase "the CORMA approach" sounds like the approach is unique to the CORMA program, but that is not the case. A more accurate phrase would be "the complete rate-matrix approach" or "the CORMA program."

- Eq 1: I could not see the definition for sigma in the main text. Although the definition does appear in the supplement, it should appear with equation 1.

- Figure 1 and the description lines 92-100: The authors compare in fig 1a and 1c the apparent rates that would result from analyzing M'(t) in comparison to the actual Solomon relaxation rate, eqn 2. It would be helpful for the reader to articulate in the figures and text "apparent relaxation rate" or somehow differentiate a rate estimated from M'(t) versus the actual rates that appear in the rate matrix, eqn 1.

- line 91: it is mentioned that Led and coworkers reported deviations in distances determined from analysis of experimental magnetization decay. Can the authors make a direct comparison of the computed results (fig 1) with the experimental data? For example, the theoretical back-calculated distances in 1b with the experimental distances calculated from the experimental longitudinal relaxation rates? The question is whether estimates calculated using the complete rate matrix analysis gives better agreement with experiment than estimates from the Solomon equation.

Typographical corrections:

- line 96: "Analogous behaviors are" should be "Analolgous behavior is"

-line 98: "differ of orders" should be "differ by orders"

-line 170: "This occurs" is missing a subject noun. Perhaps "This contribution occurs" or "This transfer occurs"

---

## Author Response (AR1)

We are glad that all reviewers like our work and we thank all of them for their constructive comments. We have addressed the points raised by them as detailed below.

**SC1**

5 *Comment 1. Interestingly, at intermediate distances the opposite effect can occur too, i.e. the total PRE can become slower than expected based on the simple Solomon equation. Is this due to NOE contacts with protons located at greater distance from the paramagnetic centre?*

As noticed, at intermediate distances the total PRE can actually become slightly slower than expected from the Solomon equation. This is indeed caused by the dipole-dipole interactions with protons at
10 larger distances from the paramagnetic metal, which cause "magnetization losses" from the closer to the farther protons. This is clarified by an additional simple simulation where 6 protons are placed along a straight line at 10, 12, 14, 16, 18 and 20 Å from a gadolinium ion. The results of the simulation are shown in the figures below. The magnetization of the proton at 10 Å recovers its equilibrium value slower than predicted from an exponential behavior, so that the monoexponential fit provides a longer
15 relaxation time. The figure also shows (black dotted lines) the magnetization curves expected from the Solomon equation for the two protons at 10 and 20 Å. Clearly the first points of the magnetization recovery for the proton at 10 Å agree with the relaxation rate predicted by the Solomon equation. This comment has been introduced in the revised version, and the figures below have been added in the Supplement (Fig. S1).

[Figure]

[Figure]

Figure S1. Calculated magnetization recovery for 6 protons placed along a straight line at 10, 12, 14, 16, 18 and 20 Å from a gadolinium ion, at 700 MHz (upper panel). The black dotted lines show the (monoexponential) behavior predicted from the Solomon equation for the two protons at 10 and 20 Å. The magnetization data calculated for the 6 protons are clearly not monoexponential. The monoexponential fits (solid colored lines) provide the relaxation rates and the back-calculated distances shown in the lower panels.

*Comment 2. Can the authors predict what the situation would be like for backbone amide protons in a perdeuterated protein in $H_2O$, where the only protons are the amide protons from the backbone and side chains (also allowing the presence of hydroxyl protons)? What would the situation be like if the methyls of isoleucine, valine and leucine are protonated whereas the rest of the protein is perdeuterated?*

We have performed the suggested calculations. As shown in the new Fig. S3A, amide and hydroxyl protons in perdeuterated conditions almost recover the rates predicted by the Solomon equation, although a slight smaller distance can anyway be calculated for protons at more than 20 Å from the paramagnetic metal. Methyl protons fully recover the Solomon behavior if the rest of the protein is perdeuterated (Fig. S3C). These plots have been added to the Supplement.

[Figure]

40

Figure S3. (A) Paramagnetic relaxation rates calculated at 500 MHz for $Cu^{2+}$-plastocyanin exchangeable (amide and hydroxyl) protons, in perdeuterated conditions. The line indicates the rates predicted with the Solomon equation. (C) Paramagnetic relaxation rates for isoleucine, leucine and valine methyl protons, assuming perdeuteration of all other hydrogens. (B and D) Agreement between
45 metal-proton distances as measured in the PDB 2GIM structure and back-calculated from the predicted $R_1$ shown in panels A and C, respectively.

*Comment 3. Line 136: exchange rates usually are expressed in s-1, not seconds.*

In the revised version we have written "with an exchange rate of $10^4$ s$^{-1}$". Thank you for having pointed this out.

**SC2**

*The CORMA program as modified by the authors clearly is very useful. Would it be possible to install the software in the NMRbox provided by Jeffrey Hoch to make it generally accessible? This would not only be helpful for others but also increase the impact and citation rate of the present article. While the original CORMA software was developed many years ago in the group of Tom James at UCSF, it may be possible to obtain permission to make the new version accessible in this way?*

We have written to the authors of CORMA and, if not forbidden, we are going to allow free access to the software from the CERM web site.

**RC2**

*-It is not fully clear from the main text what the authors mean by "calculated rates". The authors use the adapted CORMA approach, they compute the evolution of the magnetization from Eq. 4. After this, one has to check Fig. S2 (not the main text) to find a fit of recovery to obtain the "calculated rate"? If so, are the recovery monoexponential? There seem to be some deviations in Figure S2.*

It is correct that the recovery is not monoexponential. However, the "calculated rates" were indeed obtained from a monoexponential fit of the magnetization curves. We have now clarified this point immediately after the first paragraph of the Results and discussion section, by stating that "although for some nuclei the magnetization recovery curves deviate from monoexponential functions as expected (see below), the relaxation rates were calculated for simplicity as the rate constants of the assumed monoexponential time-dependences of the magnetization curves".

*-In the discussion of Figure 1, the authors should make it more clear whether the effect discussed is due to cross-relaxation with faster relaxing protons (a "selective T_1" effect) and not simply the contribution of nucleus dipole-nucleus dipole interactions to relaxation (a "non-selective T1" effect).*

Both indicated effects are effective when a non-selective pulse is applied, as done in our simulations. Most of the deviations is due to the different contributions from nucleus dipole-nucleus dipole interactions caused by the hyperfine coupling, but also cross relaxation effects are present. This can be checked by comparing the calculated rates from selective and non-selective experiments (see Figure below). This has been now clarified in the text (pag. 4).

[Figure]

Agreement between metal-proton distances in $Cu^{2+}$-plastocyanin as measured in the PDB 2GIM structure and back-calculated from the predicted $R_1$ at 500 MHz: left panel, the relaxation rates are predicted by simulating a non-selective experiment (same plot shown in Figure 1 of the manuscript), right panel, the relaxation rates are predicted by simulating a selective experiment.

*-In the light of the discussion between the authors and Gottfried Otting on a decrease of paramagnetic relaxation enhancements at intermediate distances, could this effect explain part of the surprising observations made by Flemming Hansen and Jens Led in their beautiful investigation of the blue copper site in Plastocyanin (JACS 2004)?*

Although this effect may slightly contribute to the experimental observations by Hansen and Led, we do not expect sizable contributions in that case. In fact, whereas the effects that we calculate are sizable at 13 Å or farther from the metal, the deviations were observed by Hansen and Led at less than 10 Å from the metal.

*-The authors rightfully refer the contribution to relaxation from interactions with electrons to the seminal work of Solomon by mentioning the "Solomon Equation". However, I find this expression possibly confusing, especially in the context of this investigation. The legacy of Ionel Solomon reached beyond the expression of relaxation in paramagnetic systems and it is widely accepted in NMR that the Solomon EquationS describe the evolution of magnetization in the presence of dipolar cross-relaxation, which is also perfectly relevant to this study. I believe that the authors should mention these Solomon Equations (introduced in the same Phys. Rev. 1955 article). It would be fantastic if the authors could*

*take this opportunity to clarify the Solomon Equation vs. Solomon Equations issue, possibly by referring to the former as the Bloombergen-Solomon Equation as some do in the literature.*

We have now clarified (pag. 2) that we call Solomon equation the widely used equation provided by Solomon for paramagnetic solutions (Solomon, 1955), although in the same work Solomon also provided the coupled equations which include cross-relaxation terms and should be taken into account for treating the case of nucleus dipole-nucleus dipole interacting spins. We prefer not to refer to the Bloembergen-Solomon equation because it includes contributions from Fermi-contact relaxation, which are not considered in this work.

*-I wonder if calculating the evolution of the magnetization would be easier with the use of the Homogeneous Master Equation (Levitt and di Bari 1992).*

This could be possible, but we have preferred a more "classical" approach, to better control all steps in the implementation of the model.

*-On lines 128-129, the authors mention that, in the absence of exchange, the bulk water relaxation rate is not altered by the interaction with the electron magnetic dipole. Is it true or is it an approximation since outer-sphere relaxation mechanisms are not the topic of this investigation?*

Contributions from outer-sphere relaxation are not considered in this work because of the large distance of the water molecules from the paramagnetic center, and thus not included into the model. This has been now clarified in the text (pag. 6).

*- I am not sure I understand exactly how the relaxation rates of the bulk calculated from the Solomon equation represented by dashed and solid lines in Figure 3 were calculated, what was included in each calculation.*

The relaxation rates at 1 T (solid lines) and 3 T (dashed lines) are calculated from the Solomon relaxation rates $R_{1,M}$ of the exchangeable protons in the absence of any cross-relaxation terms, according to the relationship $R_{1bulk} = R_{1dia} + f(R_{1M}^{-1} + \tau_M)^{-1}$. This has been clarified in the caption of figure 3.

*-In the spirit of Magnetic Resonance, it would be preferable that the authors publish the simulation code used in their study.*

We have written to the authors of CORMA and, if not forbidden, we are going to allow free access to the software from the CERM web site.

Minor:

*-On lines 32-34, the question of non-monoexponential evolution of polarizationin the presence of cross-relaxation could fill out volumes. The reference to the work of Banci and Luchinat is perfectly relevant*

*here but could be accompanied by a general reference to nuclear Overhauser effects (e.g. the already*
*cited Solomon article or the textbook by Neuhaus and Williamson).*

We agree to cite these more general references.

*-Similarly, the "Furthermore" on line 33 could be replaced by "Indeed" or any other suggestion by the*
*authors since cross-relaxation is the cause of non-monoexponentiality.*

OK

*-Including chemical exchange in the CORMA approach has been done in the past, for instance by*
*Jayalakshmi and Rama Krishna (JMR 2002).*

This work is now cited. Thank you for pointing it out.

*-Many symbols are not defined in the main text. I may have missed some but could not find the*
*definition of rho_i, k_i, sigma_ij, g_e, mu_B, B_0, gamma_i, tau_e, Delta_1, and tau_nu.*

All these symbols are now defined (pag. 3).

*-Unless this is a format requirement of Magnetic Resonance, I would suggest the authors use a first*
*page with article title and authors list in the supplementary information document.*

OK

**RC3**

*I have a few simple suggestions for the authors.*
*- Line 37: There were a number of groups who examined multiple-spin effects on cross-relaxation rates*
*using a complete rate matrix at the same time as Borgias and James, 1989. It would be nice to reference*
*some of the other papers as well (e.g. Boelens and Kaptein, J Mag Res 1989; Olejniczak and Fesik J*
*Mag Res 1986; co-workers and Gorenstein JACS 1990).*

These references have been added as suggested.

*- Line 55 and elsewhere: The use of "the CORMA approach" sounds like the approach is unique to the*
*CORMA program, but that is somewhat misleading as other programs from other research groups*
*utilize a complete relaxation rate-matrix approach. A more accurate phrase would be "the CORMA*
*program" or "the complete rate-matrix approach."*

The expression has been changed as suggested (complete relaxation rate-matrix approach)

*- Eq 1: I could not see the definition for sigma in the main text. Although the definition does appear in the supplement, it should appear with equation 1.*

Rho and sigma are now defined immediately after Eq. 1.

*- Figure 1 and the description lines 92-100: The authors compare in fig 1a and 1c the apparent rates that would result from analyzing M'(t) in comparison to the actual Solomon relaxation rate, eqn 2. It would be helpful for the reader to articulate in the figures and text "apparent relaxation rate" or somehow differentiate a rate estimated from M'(t) versus the actual rates that appear in the rate matrix, eqn 1.*

The expression "apparent relaxation rates" is now defined and used in the text and in the caption of Fig. 1.

*- line 91: it is mentioned that Led and coworkers reported deviations in distances determined from analysis of experimental magnetization decay. Can the authors make a direct comparison of the computed results (fig 1) with the experimental data? For example, the theoretical back-calculated distances in 1b with the experimental distances calculated from the experimental longitudinal relaxation rates?*

As suggested, we have now included in Fig. 1B the distances back calculated from the experimental rates collected by Led and coworkers for a direct comparison with computed data.

*Typographical corrections:*
*- line 96: "Analogous behaviors are" should be "Analogous behavior is"*
*-line 98: "differ of orders" should be "differ by orders"*
*-line 170: "This occurs" is missing a subject noun. Perhaps "This contribution occurs" or "This transfer occurs"*

Corrected, thank you!

[revised manuscript text omitted]

Figure S1 shows the results of calculations for a system of six protons placed along a straight line at 10, 12, 14, 16, 18 and 20 Å from a gadolinium ion. The figure also shows (black dotted lines) the magnetization curves expected from the Solomon equation for the two protons at 10 and 20 Å.  Clearly only the first points of the magnetization recovery for the proton at 10 Å agree with the relaxation rate predicted by the Solomon equation. In fact, the magnetization of the proton at 10 Å recovers its equilibrium value slower than predicted from an exponential behavior, so that the monoexponential fit provides a longer relaxation time. On the contrary, the magnetization recovery of the protons at the largest distances is steeper than predicted from an exponential behavior and the relaxation rates are sizably larger than predicted from the Solomon equation.

[Figure]

Figure S1. Calculated magnetization recovery for six protons placed along a straight line at 10, 12, 14, 16, 18 and 20 Å from a gadolinium ion, at 700 MHz (upper panel). The black dotted lines show the (monoexponential) behavior predicted from the Solomon equation for the two protons at 10 and 20 Å. The magnetization data calculated for the 6 protons are clearly not monoexponential. The monoexponential fits (solid colored lines) provide the relaxation rates and the back-calculated distances shown in the lower panels.

[Figure]

Figure S2. (left panels) Paramagnetic relaxation rates calculated at 700 MHz for MMP-12 protons, with a reorientation time of 12 ns, in the presence of high spin cobalt(II) (with an electron relaxation rate of 10 ps), copper(II) (with an electron relaxation rate of 0.17 ns), or gadolinium(III) ions (with an electron relaxation rate of 1 μs), replaced to the catalytic zinc(II) ion. The lines indicate the rates predicted with the Solomon equation. (right panels) Agreement between metal-proton distances as measured in the PDB 5LAB structure and back-calculated from the predicted $R_1$.

[Figure]

Figure S3. (A) Paramagnetic relaxation rates calculated at 500 MHz for $Cu^{2+}$-plastocyanin exchangeable (amide and hydroxyl) protons, in perdeuterated conditions. The line indicates the rates predicted with the Solomon equation. (C) Paramagnetic relaxation rates for isoleucine, leucine and valine methyl protons, assuming perdeuteration of all other hydrogens. (B and D) Agreement between metal-proton distances as measured in the PDB 2GIM structure and back-calculated from the predicted $R_1$ shown in panels A and C, respectively.

[Figure]

Figure S4. Magnetization recovery after a 90° pulse for protons at 3.0, 4.8, 8.0, 10.9, 13.2 and 15.5 Å from a $Gd^{3+}$ ion, with electron relaxation time of 36 ns, in a macromolecule with a reorientation time of 500 ns, at 3 T. The lines indicate the monoexponential fit of the data.

[Figure]

Figure S5. Relaxation rates calculated at 1 T for protons at different distance from a $Gd^{3+}$ ion with electron relaxation time of 17 ns in the macromolecular model with reorientation time of 50, 500 or 5000 ns. The lines indicate the Solomon relaxation rates calculated for the same reorientation times (colored accordingly). The relaxation rate is calculated assuming $\Delta_t$ =0.015 $cm^{-1}$ and $\tau_v$= 20 ps, instead of $\Delta_t$ =0.030 $cm^{-1}$ and $\tau_v$= 20 ps, that provide 4.2 ns at 1 T.

[Figure]

Figure S6. Bulk water proton relaxation rates calculated at 1 and 3 T as a function of the reorientation time of the macromolecular model (at 0.001 mol dm$^{-3}$ concentration) with a $S =$ 1/2 ion and electron relaxation time of 4 ns, with 100 surface protons with exchange rate of 0.1 ms. The bulk water proton relaxation rates calculated with the Solomon equation at 1 and 3 T are shown as solid and dashed lines, respectively. In all calculations, an intrinsic diamagnetic rate of 0.3 s$^{-1}$ is assumed.

[Figure]

Figure S7. (A) Half-spherical structural model with protons in octahedral geometry and the metal ion (red sphere) in the center. (B) Bulk water proton relaxation rates at 1 T as a function of the reorientation time of a macromolecular sphere containing a $Gd^{3+}$-ion in the center and protons at distances of 2, 2.5 or 3 Å (see panel A), with 100 surface protons with exchange rate of 0.1 ms. The bulk water proton relaxation rates calculated with the Solomon equation are shown as lines. In all calculations, an intrinsic diamagnetic rate of 0.3 $s^{-1}$ is assumed. The electron relaxation time of gadolinium is calculated assuming the typical values for the electron relaxation parameters, $\Delta_t = 0.030$ $cm^{-1}$ and $\tau_v = 20$ ps.